AI detectors are poor western blot classifiers: a study of accuracy and predictive values

http://orcid.org/0000-0003-1716-6210 Gosselin Romain-Daniel romain-daniel.gosselin@chuv.ch
Precision Medicine Unit, Biomedical Data Science Center, Lausanne University Hospital (CHUV) , Lausanne , Switzerland
Uversky Vladimir
Electronic publication date: 2025 Feb 20
Publication date: 2025
Volume: 13
Electronic Location ID: e18988
Received 2024 Oct 21; Accepted 2025 Jan 22
Copyright: © 2025 Gosselin
Copyright year: 2025
Copyright holder: Gosselin
License: This is an open access article distributed under the terms of the Creative Commons Attribution License, which permits unrestricted use, distribution, reproduction and adaptation in any medium and for any purpose provided that it is properly attributed. For attribution, the original author(s), title, publication source (PeerJ) and either DOI or URL of the article must be cited.
License URL: https://creativecommons.org/licenses/by/4.0/

Keywords: Research integrity, AI detection, Paper mills, Fraud, Research ethics, Accuracy study, Immunoblotting

Funding: The authors received no funding for this work.

==============================
The recent rise of generative artificial intelligence (AI) capable of creating scientific images presents a challenge in the fight against academic fraud. This study evaluates the efficacy of three free web-based AI detectors in identifying AI-generated images of western blots, which is a very common technique in biology. We tested these detectors on AI-generated western blot images (n = 48, created using ChatGPT 4) and on authentic western blots (n = 48, from articles published before the rise of generative AI). Each detector returned a very different sensitivity (Is It AI?: 0.9583; Hive Moderation: 0.1875; and Illuminarty: 0.7083) and specificity (Is It AI?: 0.5417; Hive Moderation: 0.8750; and Illuminarty: 0.4167), and the predicted positive predictive value (PPV) for each was low. This suggests significant challenges in confidently determining image authenticity based solely on the current free AI detectors. Reducing the size of western blots reduced the sensitivity, increased the specificity, and did not markedly affect the accuracy of the three detectors, and only slightly improved the PPV of one detector (Is It AI?). These findings highlight the risks of relying on generic, freely available detectors that lack sufficient reliability, and demonstrate the urgent need for more robust detectors that are specifically trained on scientific contents such as western blot images.

Introduction

In recent years, the proliferation of artificial intelligence (AI) has introduced both unprecedented opportunities and significant challenges within the landscape of academic publishing. The emergence and fast popularisation of so-called generative AI (GenAI) such as Chat Generative Pre-trained Transformer (ChatGPT) that can generate virtually any research-relevant content—such as the text of an entire article from simple textual prompting of large language models—might help authors and editors (Gruda, 2024). On the other hand, numerous dissenting voices have been raised to increase awareness about various issues linked to the use of GenAI, such as authorship, plagiarism, and reliability problems (Anders, 2023; Dergaa et al., 2023; Elali & Rachid, 2023). Beyond its ability to generate texts, the capacity of GenAI to produce virtually any content related to scientific research, such as images that are undetectable to the human eye, adds further anxiety about its possible fraudulent use to produce fake articles based on no existing data. In this context, one threat posed by GenAI is that it may further increase the activity of paper mills (Liverpool, 2023), which are potentially criminal for-profit companies that sell scientific manuscripts on demand and which have, disturbingly, been growing for years (Christopher, 2021; Candal-Pedreira et al., 2022). In the absence of easily available specific models designed and trained to detect AI-generated scientific images, professionals in the publishing sector might rely on generic AI detectors that are already accessible on the Internet, whose efficiency at spotting fabricated scientific images is unknown.

In this study, we evaluate the performance of free web-based AI detectors in identifying AI-generated scientific images. We use the example of western blotting, which is a technique often found in papers created by paper mills (Christopher, 2021). This staple technique, used in a wide range of biomedical disciplines, is employed to detect specific proteins within a biological specimen. The output of western blotting is an image (a western blot) that shows bands with patterns and intensities that provide qualitative and quantitative information for a target protein within specimens. Consistent with our unpublished pilot data, researchers have reported that realistic western blots can be easily created by ChatGPT (Zhu et al., 2024).

We selected three popular detectors (Is It AI?, Hive Moderation, and Illuminarty) and used them to analyse a dataset comprising 48 AI-generated western blot images created using ChatGPT-4 DALLE-3 and 48 genuine western blots sourced from scientific publications in 2015 (which was years before the surge of GenAI) within four biological journals. The primary aim was to estimate the sensitivity (the proportion of AI images correctly identified as AI-generated images) and specificity (the proportion of authentic images correctly identified as authentic) of these detectors and to construct confusion matrices, which are tables that summarise the performance of the detectors by displaying the number of accurate (true positive (TP) and true negative (TN)) and inaccurate (false positive (FP) and false negative (FN)) instances of image classifications. Finally, we calculated the positive predictive value (PPV; the proportion of positive hits that are indeed AI-generated) and the negative predictive value (NPV; the proportion of negative hits that are indeed authentic) of the detectors across varying prevalence rates of AI-generated images. These metrics are crucial for understanding the reliability of a detector in practical scenarios where the prevalence of AI-generated images may vary.

Our analysis reveals an important inconsistency in performance among the three evaluated AI detectors; in particular, there was a very low PPV at a realistic prevalence of AI-generated images. These results suggest that free AI-detection tools should be used with great caution when evaluating western blot images as part of peer review or editorial decisions, as their reliability for this purpose remains unproven. More specific detectors that are trained on western blot images must be urgently developed and made available to publishers and academics.

Materials and Methods

Sample size determination

The number of western blot images included in the accuracy study was determined by a sample size calculation based on the following formula (construction of an interval based on the normal approximation when the classification status is known at the time of sampling) (Hajian-Tilaki, 2014):

n=Z(1−α/2)2∗S∗(1−S)M2

where Z is the standard normal value at 1−α/2, S is the anticipated sensitivity (or specificity), and M is the maximal margin of error. A crude pilot investigation showed that the sensitivity and specificity of online AI detectors are relatively low when trying to identify western blots; they detected fake images about half of the time. With Z = 1.96, a predicted sensitivity of 0.6, and a margin of error set at 0.1, the calculation gives an estimated sample size of 92. In the absence of more precise information about the actual sensitivity and specificity of AI detectors, it was decided to evenly balance the number of AI-generated images and authentic western blot images (46/46) in the final library, thus giving a prevalence of 0.5 for the feature to be detected (i.e., a fake image). We decided to increase the sample size to 48 in each group to account for potentially unusable images.

Statistical analysis

Analyses were performed using RStudio (version 2023.06.2 + 561; RStudio Team, 2023) or GraphPad Prism (version 10) as specified in the openly provided files. Graphics were made with GraphPad Prism. The confidence level was set at 95%, which corresponds to a false positive risk (type I error) of 0.05 (i.e., 5%). The sensitivity (the proportion of AI-generated images that are correctly categorised), specificity (the proportion of authentic western blots categorised as fake), and accuracy (the proportion of correctly categorised images) were calculated for each AI detector by using Microsoft Excel for Mac (v. 16.77.1) and confirmed using the caret package in R using the counts of detector outcomes as follows:

Sensitivity=CorrectlycategorisedAI−generatedblotsCorrectlycategorisedAI−generatedblots+Falselyauthenticblots

=TPTP+FN

Specificity=CorrectlycategorisedauthenticblotsCorrectlycategorisedauthenticblots+FalselyAI−categorisedblots

=TNTN+FP

Accuracy=(0.5∗Sensitivity)+(0.5∗Specificity)

where TP, FN, TN, and FP indicate counts of true positive, false negative, true negative and false positive results, respectively. PPV and NPV were calculated for each AI detector in Microsoft Excel for Mac (v. 16.77.1) as follows, using an increasing AI prevalence (from 0 to 0.5):

PPV=Sensitivity∗Prevalence(Sensitivity∗Prevalence)+[(1−Specificity)∗(1−Prevalence)]

NPV=Sensitivity∗(1−Prevalence)[(1−Sensitivity)∗Prevalence]+[Specificity∗(1−Prevalence)]

The AI probabilities are given to one decimal place, whereas the sensitivity, specificity, accuracy, PPV, NPV, and the area under the receiver operating characteristic curve (ROC AUC) are reported to four decimal places.

Generation of fake western blot images

Fake western blot images were generated with ChatGPT 4 (https://chatgpt.com), which uses the DALLE-3 interface. Prompts, chats, and image collection were done between 20 May and 23 May 2024. The query method was based on pilot tests that evaluated the efficacy of ChatGPT 4 at creating western blots. We used repeated prompts asking for a ‘realistic image of a western blot’ while progressively changing the query to get new images, each time trying to guide the algorithm to a realistic western blot image. The entire prompting history was documented and saved. Every realistic western blot image from which four distinct bands from different lanes could be isolated was saved and used for pre-processing. Images that were not satisfactory were also saved for documentation. Four distinct chats were used to create 10–15 images each. The WEBP images created by ChatGPT were converted to PDF files to reproduce the initial format of authentic western blot images sampled from articles.

From a single query, the number of western blot bands displayed on the images generated by ChatGPT 4/DALLE-3 can vary greatly. To our knowledge, this fickleness is not, controllable by the prompt, apparently because ChatGPT does not easily correctly identify the term ‘band’ or ‘lane’. However, the number of bands in a western blot might influence the classification by AI detectors. Therefore, the choice was made to standardise the images by cropping them (by selective screen capturing, in the PNG format) to keep only four lanes, with no space to the left of the left-most band or the right of the right-most band, and leaving a space equivalent to one tenth of the width of a lane above and below the bands (see Fig. 1). To preserve independence between images and to reduce intraclass correlation, only one cropped image was taken from each AI-generated image. To investigate how the number of lanes in an image affected detector performance, we created a second dataset by isolating just two lanes from the original images. This was achieved by cropping each image to retain only the two centremost lanes.

Figure 1 General design of the study.

Western blots generated by AI (left, n = 48) were created using ChatGPT 4 and the downloaded images (WEBP format) were converted to PDF before being cropped by selective screen capturing to keep only individual bands on four lanes and saved as a PNG. Authentic western blot images (right, n = 48) were sampled from downloaded articles in articles published in 2025 in four journals. Images were obtained by selective screen capturing of individual bands on four lanes. Both AI and authentic images of western blots with two lanes were obtained by cropping (selective screen capturing, red insert) the four-lane blots. All images were scanned using three online AI detectors. The AI probability obtained for each image was both reported and used to classify them in the confusion matrix as true or false positives or negatives, and to calculate detector performances.

Sampling of authentic western blot images

Authentic western blot images were sampled from published articles. To mitigate the risk of collecting AI-generated images, photos were collected from articles published in 2015, which was before the rise of GenAI content that has been observed since 2020 (https://www.wipo.int/web-publications/patent-landscape-report-generative-artificial-intelligence-genai/en/introduction.html). Images were sampled from four journals that frequently publish western blot figures: The Journal of Biological Chemistry (electronic International Standard Serial Number (ISSN) 1559-1182, 12 articles);

Oncogene (electronic ISSN 1476-5594, 12 articles);

Public Library of Science (PLOS) Biology (electronic ISSN 1545-7885, 12 articles);

Cancer Research (electronic ISSN 1538-7445, 12 articles).

A systematic sampling scheme was used to collect the articles as follows. The final 2015 issue of each journal was examined, and each article with a figure containing a western blot image was sampled (PDF file). Only one image was sampled from each article, and only blots with one band per lane that is seen distinctly within four adjacent lanes was sampled. If the figures showed the detection of multiple proteins, then priority was given to the housekeeping protein (e.g., actin, tubulin); if no housekeeping protein was displayed, then blots of the first probed protein (scanning from top to bottom) with a single band was sampled. If multiple western blot figures were present in one article, the first western blot appearing in the article that fulfilled the aforementioned conditions was sampled. This sampling method was applied to suitable western blot images from consecutive articles, chosen while moving from the beginning to the end of the issues, until the required number of images had been obtained.

The following exclusion criteria were predefined and applied: Blots from immunoprecipitation or pull-down assays (because they might have specific background or signal intensity);

Blots either shown in colour or displaying white bands against a dark background (although these are the result of natural luminescent image acquisition, they are colour-reversed images compared with convention);

Images with less than four lanes;

Images with inserts such as a highlighted area, framed zones, arrows, or text;

Conference abstracts, reviews, or perspectives articles.

Authentic western blot images were captured using selective screenshots, and the same protocol was employed as for AI-generated blots (i.e., keeping only four lanes, leaving no space to the left and right of the bands, a space corresponding to one tenth of a lane width above and below the bands, and generating another image by cropping the photograph to keep the two central lanes).

Selection and performance analysis of AI detectors on Western blots displaying four or two lanes

A Google search using the query ‘Free AI image detector’ was performed on 17 May 2024 in Lausanne (Switzerland). The first three results that corresponded to free websites that did not require a subscription were used: Is It AI? (https://isitai.com/ai-image-detector/), Hive Moderation (https://hivemoderation.com/ai-generated-content-detection), and Illuminarty (https://app.illuminarty.ai/). Each image was scanned with all three of the AI detectors. The output of the AI detectors is a probability, given as a percentage, that the image is AI generated. Each image was classified as a TP, a TN, an FP, or an FN, with positivity defined as a detector output probability above 50%. In addition, the PPV and NPV were calculated for each detector. Two detector outputs were included in the analysis: (1) image classification determined by the detector outputs, which were used in the confusion matrix to calculate the sensitivity, specificity, accuracy, PPV, and NPV; and (2) the absolute AI probability returned by the detectors.

Data storage and sharing

AI images were saved in the WEBP format generated by ChatGPT, converted to the PDF format, and then cropped screenshot images in the PNG format were created and stored for analysis. Authentic western blot images were saved, processed, stored, and analysed in the PNG format from screenshots; no digital modifications, such as contrast, were applied to the images. All data were saved, stored, and shared on a publicly available Figshare repository (https://figshare.com).

A data folder accessible at https://doi.org/10.6084/m9.figshare.26300464 contains: The entire prompting history used to create the images;

An Excel spreadsheet that summarises all quantitative analyses;

An Excel spreadsheet that provides details of all sampled articles;

Comma-separated value (CSV) files for each specific data set;

The R codes used to analyse the data;

GraphPad Prism files used to generate the figures.

A data folder accessible at https://doi.org/10.6084/m9.figshare.26300515 contains: The cropped versions of the authentic western blots;

The full AI-generated images and their cropped versions;

The unused (failed) ChatGPT 4 images.

A preprint version of this study is available at https://doi.org/10.48550/arXiv.2407.10308.

Results

AI detectors showed highly variable abilities to distinguish between artificial and authentic western blots

Following the assessment of the western blots with four lanes, the returned AI probabilities spanned a very wide range for each AI detector (Fig. 2A). Illuminarty gave the highest average AI probabilities (median = 86.2, interquartile range (IQR) [42.2–98.7] for AI-generated western blots, median = 81.1, IQR [19.2–99.1] for authentic western blots). The lowest AI probabilities were returned by Hive Moderation (median = 17.0, IQR [9.5–30.3] for AI-generated western blots, median = 18.2, IQR [8.3–37.0] for authentic western blots). Is It AI? produced an intermediate output that was visibly dissimilar between AI-generated western blots (median = 85.1, IQR [69.8–93.1]) and authentic western blots (median = 47.3, IQR [26.9–66.0]). This variability in detector output was reflected in the formal confusion matrices and accuracy analyses presented in Fig. 3B. The proportion of false positives was 22/48 for Is It AI?, 6/48 for Hive Moderation, and 28/48 for Illuminarty, and the proportion of false negatives was 2/48 for Is It AI?, 39/48 for Hive Moderation, and 14/48 for Illuminarty. As presented in Fig. 4A, the consequence of these high rates of misclassifications is that the performance of the detectors often fell short of usual standards (Is It AI?: sensitivity = 0.9583, specificity = 0.5417, accuracy = 0.7500, ROC AUC = 0.9028, 95% CI [0.8435–0.9621]; Hive Moderation: sensitivity = 0.1875, specificity = 0.8750, accuracy = 0.5313, ROC AUC = 0.5100, 95% CI [0.3930–0.6270]; Illuminarty: sensitivity = 0.7083, specificity = 0.4167, accuracy = 0.5625, ROC AUC = 0.5449, 95% CI [0.4276–0.6622]).

Figure 2 Distribution of AI probability returned by AI detectors.

The violin plots show the density distribution of AI probabilities (given as percentage on y-axis), with individual data points shown (each dot represents a single western blot image). For each AI detector (IS It AI? in blue, Hive Moderation in green, and Illuminarty in red), the group of AI-generated western blots is shown on the left (n = 48) and the authentic western blots is shown on the right (n = 48). (A) Four-lane western blots. (B) Two-lane western blots. The horizontal dotted line indicates the limit (50% probability) set to define positive results.

Figure 3 Confusion matrices.

The confusion matrices show the quality of the classification systems. For each table, the genuine status of the images is given on the left-hand side and the status given by the detector is shown on the top. The four possible outcomes of the 2 × 2 matrices are shown in (A). TP, true positives; FP, false positives; FN, false negatives; TN, true negatives. (B) Counts of different outcomes for western blots with four lanes. (C) Counts of different outcomes for western blots with two lanes.

Figure 4 Performance and receiver operating curves (ROC) of AI detectors.

The tables on the left show the sensitivity, specificity, and accuracy of each AI detector. On the right, ROC curves are given for each detector (IS It AI? In blue, Hive Moderation in green, Illuminarty in red). The area under the curve (AUC) is given as a point estimate with 95% confidence interval between brackets. (A) Matrices and ROC obtained from western blots with four lanes. (B) Matrices and ROC obtained from western blots with two lanes.

As shown in Fig. 2B, when the scanned western blots were cropped from four lanes to two lanes, the detectors returned markedly reduced AI probabilities (Is It AI?: median = 62.8, IQR [38.5–80.8] for AI-generated western blots, median = 11.7, IQR [6.2–28.8] for authentic western blots; Hive Moderation: median = 6.2, IQR [3.1–18.0] for AI-generated western blots, median = 12.2, IQR [5.6–24.1] for authentic western blots); Illuminarty: median = 15.4, IQR [7.8–43.5] for AI-generated western blots, median = 9.1, IQR [2.4–29.0] for authentic western blots). Consequently, there was a dramatic increase in the number of false negatives (Is It AI?: 20/48; Hive Moderation: 46/48; Illuminarty: 38/48) and a reduction in the number of false positives (Is It AI?: 4/48, Hive Moderation: 4/48, Illuminarty: 8/48; Fig. 3C). Figure 4B shows that although the overall test accuracies were not noticeably impacted by reducing the number of bands (Is It AI?: accuracy = 0.7500, ROC AUC = 0.8974, 95% CI [0.8361–0.9586]; Hive Moderation: accuracy = 0.4792, ROC AUC = 0.6252, 95% CI [0.5118–0.7386]; Illuminarty: accuracy = 0.5208, ROC AUC = 0.6248, 95% CI [0.5115–0.7380]), there was reduced sensitivity (Is It AI?: 0.5833; Hive Moderation: 0.0417; Illuminarty: 0.2083) along with increased specificity (Is It AI?: 0.9167; Hive Moderation: 0.9167; Illuminarty: 0.8333).

Evaluation of the PPVs and NPVs of the AI detectors

Sensitivity and specificity are intrinsic properties of the detectors that describe their ability to correctly classify images as authentic or fake. However, in editorial contexts where the question is whether a given image can be deemed authentic, the PPV and NPV would be more useful; these measures are functions of both sensitivity and specificity, as well as the prevalence of AI-generated images in the literature. Therefore, we calculated the PPV and NPV for each of the three AI detectors in scenarios in which AI prevalence increased sequentially from 0 to 0.5 (Fig. 5). The data from western blots with four lanes (Fig. 5A) showed that the PPV of each detector was very low when the AI prevalence was set below 0.1, with a maximum value of 0.1885 for Is It AI? (0.1885) when the AI prevalence was set at 0.1. This indicates that most of the western blots categorised as AI generated would be false positives. When further reducing the AI prevalence in the simulations, the PPV become dramatically low. For example, at a prevalence of 0.005 (meaning that we expect one western blot out of 200 to be AI generated), the PPV was 0.0104 for Is It AI?, 0.0075 for Hive Moderation, and 0.0061 for Illuminarty. Conversely, the NPV was greater than 0.9 for all detectors at this realistic AI prevalence, indicating that false negatives would be rare in this scenario. At higher AI prevalences, the PPV for each detector increased steadily as the NPV decreased, although a discrepancy was observed between the relatively high NPV of Is It AI? (NPV = 0.9285 at an AI probability of 0.5) and the NPV of Hive Moderation (NPV = 0.5185 at an AI probability of 0.5) and Illuminarty (NPV = 0.5185 at an AI probability of 0.5). The PPVs and NPVs calculated using western blots with two lanes (Fig. 5B) showed patterns for Hive Moderation and Illuminarty that were similar to those obtained from four-lane blots. However, Is It AI? showed a higher PPV than that of the two other detectors, even for low AI prevalences, and a lower NPV than when blots with four lanes were analysed.

Figure 5 Positive and negative predictive values.

The graphs show the positive predictive value (PPV, left) and negative predictive value (NPV, right) calculated for different theoretical prevalences of AI-generated images between 0 and 0.5. (A) Predictive values obtained from western blots with four lanes. (B) Predictive values obtained from western blots with two lanes. IS It AI? in blue, Hive Moderation in green, Illuminarty in red.

Discussion

The present study has addressed a critical issue about the efficacy of current free AI detectors in recognising AI-based image forgery in scientific publications. Using western blots as an example, we demonstrated that none of the sampled AI detectors would be adequate for aiding editorial vigilance in detecting images created by GenAI. It remains uncertain whether AI-generated western blots have already infiltrated the biological literature. Our findings that free detectors show different performances and often misclassify western blot images largely align with previous studies on AI-generated texts (Bellini et al., 2024; Flitcroft et al., 2024; Howard et al., 2024; Odri & Ji Yun Yoon, 2023; Pan & Florian-Rodriguez, 2024; Popkov & Barrett, 2024). This is particularly concerning because reliable automated detection methods are critically needed, as advancing technology makes AI-generated scientific images progressively more difficult for humans to spot (Hartung et al., 2024; Wang et al., 2022). Recent efforts by others to develop specialised western blot classifiers are promising (Wang et al., 2022; Mandelli et al., 2022), and our results strongly support the need for further research in this direction. Upon completion, these algorithms should be made freely accessible to journal editors and researchers to help safeguard scientific integrity.

One novel finding is the large variation in sensitivity and specificity values that was observed across the three tested detectors. The AI detectors often generated completely opposing AI probability outputs for a given image. This emphasises the importance of the different mathematical architectures on which the various tools are based. Beyond their inconsistency, the values obtained for these metrics often fell well below the acceptable thresholds for reliable detection, indicating frequent image misclassification. This result is in clear contrast to the high sensitivity and specificity reported for the detection of AI-generated text (Gao et al., 2023; Miller et al., 2023), demonstrating the relative immaturity of free AI image detectors. When examining the PPV and NPV, which would be the essential measures for editorial decision-making, the results indicated that the PPV was low at realistic probabilities of AI-generated images in the literature. This suggests that concluding about the falseness of a western blot image based on a high AI probability given by an AI detector would often be misleading. The actual prevalence of AI-generated images in published articles is currently unknown, and it is challenging to estimate in the absence of reliable detectors. Nevertheless, the existing estimates indicate that the prevalence of text created by AI in publications already exceeds 10–30% (Howard et al., 2024; Miller et al., 2023; Bisi et al., 2023; Pesante, Mauffrey & Parry, 2024), and the rate of inappropriate image manipulation is projected to be 5–30% (Heathers, 2024). These figures suggest that AI-generated images might already be well entrenched in the scientific literature. Should that be the case, they would likely be the products of earlier, presumably less sophisticated generative algorithms than those tested in this study, and the question that arises is whether the current AI detectors could be more effective at identifying these older cases.

The binary nature of western blots—black bands on a white background—combined with their low resolution and frequent tight cropping provides minimal data points for algorithmic analysis, making them particularly challenging for manipulation detection. Therefore, it is reasonable to believe that western blots will be among the first scientific digital images to be doctored using AI. Notably, our results also indicate that reducing the number of lanes in the images resulted in decreased detector sensitivity and increased specificity. This relationship between western blot complexity in terms of band richness and detector output will have to be accounted for when integrating AI detection in the editorial process. The fact that image classification may be affected by image editing further highlights the critical importance of thoroughly documenting and providing data for all steps of image pre-processing, as well as the importance of systematically sharing unprocessed raw images.

One way to help improve the detection process could be to use automated AI detection as a subsequent step following a human-led preliminary identification process that is applied only to suspicious articles. This approach mirrors the traditional distinction in epidemiology between large-scale screening tests that are performed agnostically vs diagnostic tests performed on symptomatic patients. Automated AI detection would be applied exclusively to articles flagged with potential AI indicators, such as authors with multiple article retractions or research fields that are known for frequent publication of AI-generated images. This two-step strategy could enhance the PPV by increasing the probability of AI in articles in the sample. The improvement of AI detection will also fundamentally hinge on the inclusion of western blot images in training sets used to develop image detectors. Improving the performance of detector algorithms on scientific images, such as western blots, requires their rigorous design and training on large collections of authentic western blot images.

The first limitation of this study pertains to the potential non-representativeness of the three tested detectors. It is imaginable that other detectors, particularly those that require a subscription, might exhibit superior performance (Popkov & Barrett, 2024). However, this proof-of-concept study strongly suggests that finding an effective free detector would currently require excessive and impractical efforts, further supporting our recommendation against their use. In relation to this first limitation, we restricted our sampling images to western blots created by ChatGPT due to the dominant popularity of this software, but images from other generators might be classified differently by AI detectors. Second, images were classified as AI generated if their AI probability was at least 50%. This threshold is expected to significantly impact the rates of false positives and false negatives, consequently affecting the calculation of performance metrics (Howard et al., 2024). Future studies should explore the impact of varying thresholds on the ability of detectors to distinguish between authentic and fake images. We used PNG images from screenshots because we had no access to raw images and because we assumed that a similar approach would be used by peer reviewers and editors. However, the image format (e.g., JPEG and TIFF) might influence the detector output. Therefore, future detector algorithms should be trained using various formats. Finally, we had no indication about whether the authentic western blots included during our sampling had been acquired using photographic film exposure or using a charge-coupled device (CCD) camera. It is possible that images obtained through different acquisition methods would give different AI probabilities when scanned by AI detectors.

Conclusions

In conclusion, this study uses the example of western blots to raise awareness about the urgent need for more effective AI detectors that are specifically designed and trained to reveal fake scientific images. If such detectors have indeed been developed, whether by academic or for-profit entities, their existence and performance metrics must be made publicly available to enable editors and reviewers to effectively screen manuscript submissions, as currently there appears to be no evidence of such specialised tools within the academic community. The implications of our findings are profound for editors, publishers, and reviewers tasked with maintaining the integrity of the scientific literature. Enhanced AI detection capabilities, coupled with rigorous editorial policies and reviewer training, are vital for upholding the standards of scientific publishing.

The author would like to thank Pr. Jacques Fellay, head of the department, for approving this research direction. AI (ChatGPT 4.0) was used to generate half of the images included in the dataset since the purpose of the study is to test the efficacy of AI detectors to detect AI-generated images.

Additional Information and Declarations

Competing Interests

The author declares that they have no competing interests.

Author Contributions

Romain-Daniel Gosselin conceived and designed the experiments, performed the experiments, analyzed the data, prepared figures and/or tables, authored or reviewed drafts of the article, and approved the final draft.

Data Availability

The following information was supplied regarding data availability:

Data are accessible at Figshare:

Gosselin, Romain (2024). Supporting data: AI Detectors are Poor Western Blot Classifiers: A Study of Accuracy and Predictive Values. figshare. Dataset. https://doi.org/10.6084/m9.figshare.26300464.v1.

Gosselin, Romain (2024). Supporting data (images): AI Detectors are Poor Western Blot Classifiers: A Study of Accuracy and Predictive Values. figshare. Dataset. https://doi.org/10.6084/m9.figshare.26300515.v1.

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
