# Peer review of "AI detectors are poor western blot classifiers: a study of accuracy and predictive values"

_PeerJ, doi:10.7717/peerj.18988_

## Round 0.1 · original submission · Minor Revisions

Please address concerns of all reviewers and revise manuscript accordingly.

Reviewer 1 ·

Basic reporting

Clear and concise. Adequate.

Experimental design

Adequate

Validity of the findings

Good. It will give an insight into the use of AI detectors for western blot images.

Additional comments

Can be considered for publication.

Reviewer 2 ·

Basic reporting

I have no issues with the language. There are some grammatical issues in places, but this never got in the way of understanding. The structure is good, figures and tables are clear, and the study is self-contained. I felt the paper was a bit light on the references to pre-existing literature on AI-generated images in scientific literature, instead focusing on AI-generated texts. I would suggest maybe looking into this a bit (see specific comment below).

Experimental design

The experimental design is clear and the methods are described clearly throughout. Only with respect to producing the two-lane western blots do I think that I would not be able to reproduce the author's methods. In every other aspect this is an A+ in that aspect. With respect to research question, I feel that the author has bigger ambitions in places (saying something generalizable about the state of detecting AI-generated images) than the methods can supported by the method (challenging three random detectors found through a simple google search).

Validity of the findings

The findings seem valid, although not as generalizable as they are made out to be. It should also be noted that the employed sample sizes are quite low, due to a fairly flimsy sample size estimate.

Nevertheless, I think this is an important study that is timely, meaningful, and of use to readers. I want to be clear on this point.

Additional comments

Gosselin has studied how three free web-based services for detecting AI-generated images perform when asked to distinguish synthetic images from real ones. The performance is uneven and not very impressive, leading the author to conclude that these services cannot currently be used to ward off a potential influx of fabricated images in journals by nefarious individuals. The study is transparent throughout, clear in its message and quite timely.

Whereas I agree mostly with the author’s perspective on this problem not necessarily being solvable through suspect web services, I worry that they are creating a strawman argument.

In the discussion segment, Gosselin discusses the limitations of the study, which is commendable. But it does not change the fact that three detectors found through a Google search are not necessarily representative for the state of detectors. When I repeat the author’s searches and follow the instructions, I come up with another set of detectors entirely. One of the three tested ones (“Is it AI?”) now requires registering, for example. But, also, no one of the detectors have been built for the task they are put through. It leaves the reader with additional questions: What if the good ones just require the user to register a profile? What if the good ones require payment? What if the fourth one on the list would have blown the other out of the water?

At no point are we given any information on what methods these detectors employ to detect AI-generated images either. Perhaps this is information that is not available, but it feels like some really important context is missing here.

Moreover, the chosen types of images – western blots – are some of the most difficult ones to scrutinize for manipulations. Black and white letterboxed images shown at a low resolution gives the tools very little to work with. And then the images are cropped down to show only two blots! Here it feels as if the authors is being quite unfair on the detectors.

At the very least, the author would need to discuss these issues, since the cards seem very much stacked against the “AI detectors” in how the study was designed.


Below, I have listed some additional comments by the relevant section.

Abstract

“This highlights the impossibility in confidently determining image authenticity based on detectors.” Just because the author could not confidently determine faked images from real does not, by necessity, make the process impossible. It may be that it is, indeed, impossible. But, the author has not proven this by any stretch.

Introduction

“...whose efficiency at spotting scientific images is unknown.” Should read “fabricated scientific images.”

“These results indicate that the AI-detecting tools that are currently available for free cannot be used...” Well, this depends on whether these three are representative of all the tools that are available at current. We would need more background on what is out there, and why these three platforms were selected. Or more careful wording.

Methods

The sample size calculation seems a bit flimsy. The margin of error is set at 10%. This can be contrasted with the author presenting sensitivity and specificity at a thousand times higher precision (!). I would also assume that these imprecise estimates could have affected the analyses shown in Figure 5 quite a lot. Was the generation of western blot images time-consuming, or why was the margin of error set so high?

The methods do not adequately describe the generation of the two-lane blots.

Are the images/prompts timestamped somehow? I could not find this in the Figshare catalogues. I imagine that ChatGTP goes through a lot of tweaks, fairly regularly. I would imagine that it could be relevant, down the road, to know when (version etc.) the images were generated.

Does the author feel that using two-lane blots give the detectors “a fair chance?” I could crop an egregiously AI-generated image, only to show a cluster of twelve pixels and no detector – computer-based, human, or otherwise – could tell it apart from twelve pixels extracted from my vacation photos. Surely, no one expects the detectors to work without enough material to assess?

Discussion

“It remains uncertain whether AI-generated Western blots have already infiltrated the biological literature, and the current detection tools would not yet be capable of efficiently identifying them.” Well, this is not necessarily true. If AI-generated images have made it into publications (highly likely in my mind), those images would have been generated using an earlier generative AI algorithm than what the author has used in this investigation. It is worth discussing whether these tools would work better in discovering older (presumably more primitive in construction) AI-generated images. This is speculation, I know. But, realistically, we are not looking to win the arms race between the ability to generate authentic-looking images using “AI” and the ability to discern these from “real” images. A more achievable target would be to see if we could reliably identify the “last generation” of AI-generated images.

“...the rate of inappropriate image duplication is projected to lie around 5% [20].” I will never disparage citing Bik et al. But this estimate is a decade old now (the analysis included papers between 1995 and 2014). A recent preprint (Heathers, 2024) lists considerably higher numbers when summarizing investigations similar to Bik et al. carried out over the last ten or so years.

“In conclusion, this study uses the example of Western blots to raise awareness about the urgent need for more effective AI detectors that are specifically designed and trained to reveal fake scientific images.” Respectfully, the author cannot say that these detectors do not already exist. Googling “free AI image detector” and selecting three top hits is not an exhaustive analysis of what is available and it is certainly not the way to check whether there are detectors specifically built to find fake scientific images.

Overall, the discussion is a bit thin on references to recent investigations into AI-generated images in the scientific domain. Mostly, the author makes comparisons between detecting AI-generated text and the present study. But Hartung et al. from earlier this year (“Experts fail to reliably detect AI-generated histological data”) seems like an obvious reference to discuss. Perhaps also L. Wang et al. from 2022 (“Deepfakes...”) and S. Wang et al. from 2020 (“CNN-generated images are surprisingly easy to spot... for now”). It is of course the author’s prerogative to use the references they see fit, but it seems like the paper avoids engaging in the discourse that is already out there.

·

Basic reporting

This study by Gosselin et al., examines the efficiency of three free web-based AI detectors (Is It AI, Hive Moderation, and Illuminarty) in identifying AI-generated Western blot (WB) images. Gosselin provide detailed methods, including sample size determination, statistical analysis, and the factors considered when generating and sampling the WB images. Tested on these detectors were 48 AI-generated images created using ChatGPT-4 and 48 genuine images from articles published in 2015. The authors found significant variation between the sensitivity and specificity of the detectors. Also, the analyses showed low positive predictive values (PPV) for the detectors. Overall, the study concludes that the current AI detectors are unreliable in accurately and correctly identifying AI-generated images.
Data significance: the authors provide important insights into the limitations of existing AI detectors for the purpose of maintaining the integrity of scientific publications. Significant aspects of the data include variability in detector performance, which emphasizes the irregularity and unreliability of current free AI detectors in identifying AI-generated Western blot images, and the discovery of a low positive predictive values (PPV), thus indicating a high rate of false positives. This means that many images classified as AI-generated by the detectors were authentic, making it difficult to confidently determine image authenticity based on these tools. Furthermore, it was shown that the image size (in terms of number of lanes) impacts the performance of the AI detectors.
Constructive criticism/suggestions:
- Lines 26-27: “This highlights the impossibility in confidently determining image authenticity based on detectors”. Is it really impossible? This statement counters the closing sentence of the Abstract where it was stated that there is a need to develop more robust detectors trained on large scientific datasets such as western blot images to be able to determine image authenticity. If it was impossible to determine, why is there a statement that indicates the possibility of assessing image authenticity on more robust AI detectors? I believe the last 3 sentences of the abstract should be reworded or rephrased.
- The introduction needs to be reworked for flow and transition. For example, “however” in line 40 should be removed because “on the other hand” is a synonym. Another example is in line 180, “Blots show in colour of with…”. Also, some words were repetitively used in some sentences. I suggest a colleague who is highly proficient in written English language review the manuscript because some of the errors found in the text make comprehension a bit challenging.
- In line 62: the 3 detectors used in the study should be mentioned/introduced.
- Line 70-71 should be reworded. “These metrics are crucial for understanding detector reliability in practical scenarios in which the prevalence of AI-generated images may vary.” This sentence may be rewritten as “These metrics are crucial for understanding a detector reliability in practical scenarios where the prevalence of AI-generated images may vary”.
- A reference to an article or page regarding the timeline of GenAI should be included in Lines 156-157. For example, https://www.wipo.int/web-publications/patent-landscape-report-generative-artificial-intelligence-genai/en/introduction.html.
- I think a brief description of what the confusion matrices (TP, FP, TN, FN) mean for the outputs should be included in the introduction.

Experimental design

see above

Validity of the findings

see above

Additional comments

see above

---

## Round 0.2 · accepted · Accept

I can confirm that in my opinion all issues pointed out by the reviewers were adequately addressed and the revised manuscript is acceptable now.